# Enhancing human aspects of care with young people with muscular dystrophy: An evaluation of a participatory qualitative study with clinicians

Jenny Setchell[1,2]*, Donya Mosleh[3], Laura McAdam[4,5], Patricia Thille[6], Thomas Abrams[7], Hugh J. McMillan[8], Bhavnita Mistry[2], Barbara E. Gibson[2,9]

1 School of Health and Rehabilitation Sciences, The University of Queensland, Brisbane, Australia,
2 Bloorview Research Institute, Holland Bloorview Kids Rehabilitation Hospital, Toronto, Canada,
3 Rehabilitation Sciences Institute, University of Toronto, Toronto, Canada, 4 Developmental Paediatrician and Clinician Investigator, Bloorview Research Institute, Holland Bloorview Kids Rehabilitation Hospital, Toronto, Canada, 5 Department of Paediatrics, Division of Developmental Paediatrics, University of Toronto, Toronto, Canada, 6 Department of Physical Therapy, University of Manitoba, Winnipeg, Canada, 7 Queens University, Kingston, Canada, 8 Children's Hospital of Eastern Ontario, University of Ottawa, Ottawa, Canada, 9 Department of Physical Therapy, University of Toronto, Toronto, Canada

* j.setchell@uq.edu.au

**Data Availability Statement:** The original approved protocols by The Holland Bloorview Research Ethics Board (REB) did not have a provision for

## Abstract

### Purpose

This paper evaluates a study which aimed to enhance clinical care of young people with Duchenne or Becker muscular dystrophy (MD) and their families in two Canadian neuro-muscular clinics. We report on how/why the study changed clinical practices in relation to the 'human' (e.g., emotional, social, existential, cultural) dimensions of living with MD.

### Materials and methods

The intervention involved regular dialogical exchanges with clinicians across the two sites, during which direct observations of the clinics' care practices were discussed and changes were planned. We drew from realist evaluation approaches to assess changes in clinical care associated with the intervention. Data sources included dialogical exchanges; clinic observations; interviews with clients, families and clinicians; and team analysis sessions.

### Results

Our evaluation suggests the clinical teams shifted their thinking and practices towards greater consideration of human aspects of living with MD including: more routinely attending to emotional, social and experiential dimensions of living with MD; reconceptualisation of risk; and considerations of affective aspects of clinical care. Not all clinicians changed their thinking and practices in the same ways, or to the same extent, and there were differences between the sites. These differences were likely due to numerous factors, including varying levels of clinician comfort with examining and shifting their own practices, and differing formal and informal clinic routines at each site.

data sharing and the study participants did not consent to sharing their data. We have confirmed with the REB Chair that we are not permitted to store the data in a repository. REB reference numbers: 15-608 & 17-729. Data access requests can be sent to researchethicsboard@hollandbloorview.ca.

**Funding:** This work was supported by a AHSC AFP Innovation Fund Grant (JS,PT,LM,BG,TA,HM); a Holland Bloorview Centres for Leadership Grant (JS,BG,LM); and an Associated Medical Services Phoenix Project Call to Caring Grant(JS,LM,BG, TA). JS is supported by a National Health and Medical Council fellowship. BG is supported by the Bloorview Kids Foundation Chair in Childhood Disability Studies.

**Competing interests:** The authors have declared that no competing interests exist.

## Conclusions

Overall, this intervention was able to shift clinic practices, and could feasibly be adapted across rehabilitation settings.

## Introduction

Duchenne and Becker muscular dystrophy (DMD/BMD) are collectively referred to as 'dystrophinopathies'; they arise from a genetic mutation resulting in the absence (DMD) or reduction (BMD) of functional dystrophin protein [1]. Since the gene is located on the X-chromosome, these forms of MD almost exclusively affect boys or men. They are progressive conditions where muscular strength declines over time, resulting in young people with DMD often using power wheelchairs by their teens [1]. Those with BMD experience a later onset and slower progression. For the purposes of this article, we shall refer to these two dystrophinopathies as 'MD'.

MD research and healthcare practice has had a considerable and persistent focus on biomedical and rehabilitation aspects of care such as finding a 'cure', increasing longevity, improving function and reducing physical risks [1–3]. At times, this has come at the cost of the 'human' aspects of care. We use the term "human" as a shorthand that extends beyond "psychosocial" concerns to capture the emotional, existential, social, and moral dimensions of illness experiences and care that co-exist with the physical/biological and technical dimensions [4, 5]. For example, human aspects include dealing with disability stigma, creating meaningful shortened lives, and managing family interactions [4, 5]. The human dimensions of living with MD overlap with and extend the concept of person-centred care, which, at least in practice, tends to focus on autonomy and decision making [6, 7]. Our conceptualisation of the human aspects co-exists with biomedical dimensions and is interrelated with them. Attending to these human dimensions aligns with arguments that extending life, or enhancing participation of people with MD in society, should not be the only foci: an enriching and enjoyable life (regardless of its length) is also highly salient.

Clinical practices are difficult to shift [8]. Even with clear parameters to change, such as applying clinical guidelines, multidisciplinary teams' attempts at change are often limited or ineffective [9]. Many factors influence the ability to change, including systemic constraints [10], cultural norms [11, 12] and the variability of what is required for change across contexts [9, 13]. In their review of primary care, Lau, Stevenson [14] stated: "Implementation of any type of intervention is complex, dynamic and influenced by a variety of factors at the level of external context, organisation, professional and intervention" (p. 37). There are deeply ingrained ways of acting, thinking and doing that are formed through repetition and internalisation of social norms [15]. These learned habits are often mistaken as inherent or natural [16], including in the context of disability healthcare and MD [17], and are thus very hard to change or even notice. Clinical practice has multiple habitual practices, values and assumptions. As a result, it is not surprising that there is a general agreement in the literature that to successfully change clinical practice, interventions need to be sustained and repeated, and be contextually relevant [14, 18, 19].

In response to these challenges, we developed a project that aimed to: 1) examine the assumptions unpinning clinical practice; 2) create adaptable recommendations for change to ensure context-specific relevance; and 3) change multiple aspects of clinical practice towards enhancing the human aspects of living with MD [5, 20]. This paper reports on the impact of

the interventional aspects of that project by responding to the following research question: How and why did the project change clinician thinking and practices in relation to the human aspects of care in outpatient neuromuscular clinics?

## Materials and methods

### Overview

In this paper, we evaluate changes to clinician thinking and practices that comprised part of a research project to examine and change MD care. The project was a 3-year collaboration between researchers, clinicians and young people with MD and their families at two neuro-muscular clinics across two large Canadian cities. Each clinic provides multidisciplinary, sub-specialty care to children and young adults (< 18 years old) with neuromuscular diseases. The focal point for conceptualising and making changes from the project was a series of 2-hour dialogical exchanges between researchers and clinicians at each site, during which observations of the clinic were analysed, and recommendations for change were co-developed and imple-mented by the clinicians. We describe these exchanges as 'dialogical' because they differed sub-stantively from 'focus group' methods; they were designed according to participatory research practices, and thus did not assume that any one perspective (clinical or research) was authori-tative or complete [20]. Our ability to foster dialogical exchange was enhanced by our own var-ious disciplinary perspectives, including disability studies, bioethics, social psychology, sociology, geography, physiotherapy and medicine, which enabled dialogue without closure among the research team as well (reviewed in more detail in Thille, Gibson [20]). Our evalua-tion was also informed by consultatory interviews with clients and their families. Details of the project methodologies have been reported elsewhere [5, 20]; these papers include early evalua-tions of the project at the first site. All work was conducted with approval of the institutional research ethics boards.

### Methodology and theoretical underpinnings

To be consistent with the critical social science underpinnings of our study, we sought evalua-tion approaches that conceptually aligned with the study. After considering a number of evalu-ative approaches, we drew from elements of a realist approach. Realist evaluation approaches (RE) consider the complexities of an intervention through a focus on improving understand-ings of *how* and *why* interventions work or do not work *in particular contexts* [21]. RE is underpinned by the premise that causation is not easily accessed within a complex system and cannot usually be directly observed, but can be investigated. RE considers both material and social worlds as 'real' in that they can have 'real effects' [21]. Material environments, clinical tools, measuring devices and clinical practices, as well as programs, policies, social institutions are considered to have an impact on how and whether an intervention works. Aligned with our critical social science approach to the project, the underlying assumption of RE is that there is no final truth/knowledge but that it is possible to work towards a closer understanding of what is happening.

   We draw in particular from the Realist Evaluation Framework which has been increasingly and diversely applied in healthcare environments [e.g., 22, 23]. Following realist principles, the framework supports an analysis of the ways in which context affects how, and for whom, an intervention works–thus acknowledging that observed outcomes are not simply the result of the intervention but of how it interacts with elements of the context [24]. Our approach drew on some of its principles (primarily its 'outcome elements' which focus on who the interven-tion works for, and how the intervention application and context affect its impact) but used a flexible design in keeping with the project goals and theoretical groundings. Our analysis was

applied post-hoc to any data collection so did not include iterative elements of a RE (e.g., no Initial Program Theory). Our evaluation approach was discussed and refined by members of the research team JS, BG, BM, PT, LM to develop evaluative questions from the 'overarching and subsidiary' questions delineated by Westhorp [21].

## The interventional components

The overarching project was undertaken to both *examine* and *understand* existing practices [5, 20, 25–28] as well as to *intervene* to change clinical practices. The *interventional* components are the focus of this paper. The key intervention strategies were a series of approximately quarterly, 2-hour meetings ("dialogues") between the researchers and participating clinicians at each site. During dialogues, the researchers facilitated the clinicians to reflect critically on their current practices and how they might be changed to enhance the human aspects of living with MD. The researchers were experienced with using critical reflexivity in their own clinical practices, and trained to facilitate critical reflexivity in practice for others. The dialogues occurred alongside a series of longitudinal ethnographic observations of clinical interactions and practices at both sites.

The reflexive processes we used included sharing theoretical concepts, co-analysing selected observation field notes, identifying opportunities for change, and assessing the changes as they were implemented in clinical practice. The dialogues (including recommendations for practice changes) were informed by ongoing discussions with the clinicians (during dialogues, clinician interviews, and frequent contact with the two site medical leads who were co-investigators on the study), and engagement with clients and families through consultations and interviews (Fig 1). For more information about the reflexivity approach underpinning the dialogues, and their dialogical nature, at one site, see [20].

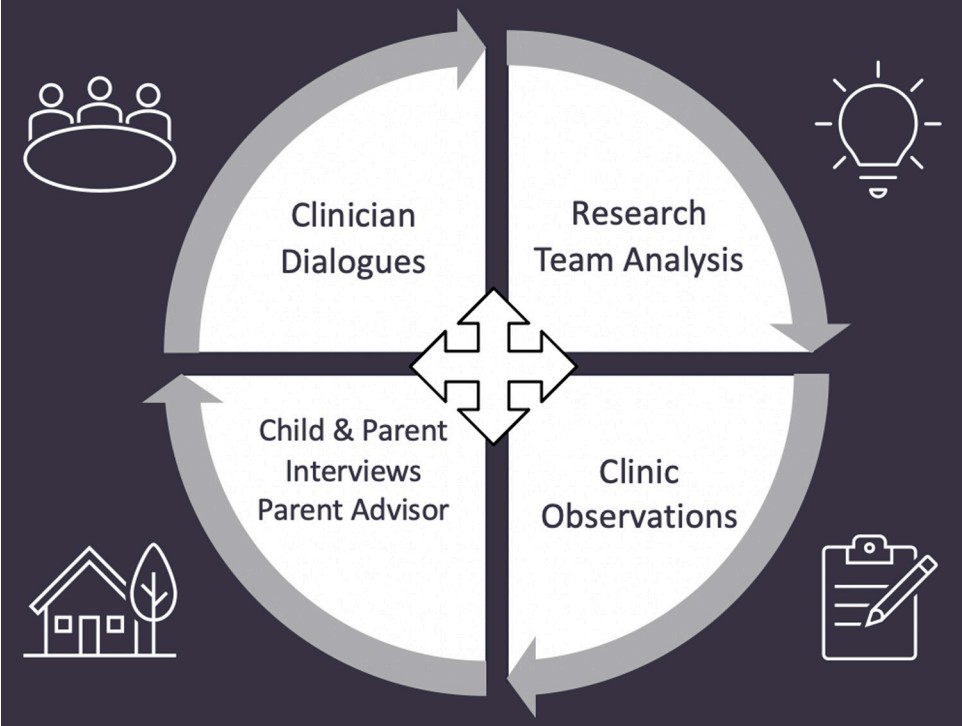

**Fig 1. Interactions between various elements of the intervention and broader study project.**

## Study sites and participants

We studied two outpatient children's neuromuscular clinics located in different Canadian cities. Although all methods were included in each site, there were considerable differences in the intervention between sites. The study duration was considerably longer at site 1 (3 years) than at site 2 (1.5 years), There were half the number of dialogues and approximately one quarter the number of observations at site 2. The different study durations were the result of our success at attracting additional funding to add a second site after work at the first site was underway. There were also existing differences in the clinical environment, including a different mix of healthcare professional disciplines. These differences, and their implications, are discussed below. Both clients/families and clinicians were participants.

**Client participants.** Clients and their families were deemed eligible if they had genetically-confirmed DMD/BMD and were willing to provide informed consent to participate. They were recruited through existing client databases and written informed consent for all patient and family participants was obtained in accordance with institutionally approved procedures.

**Clinician participants.** All clinicians at each site were invited to provide consent to participate in the study during a site information session, or contacted individually if they joined the clinics after study commencement.

## Data sources

Congruent with a realist approach [21], we analysed numerous data sources produced from the project to determine the impact of the intervention across the two sites (Table 1). Most data were qualitative, including the detailed observation field notes of clinic visits (site 1: 36, site 2: 8, each 2–4 hours in duration); summaries and/or transcribed audio recordings of dialogues (site 1: 10, site 2: 5); transcribed audio-recordings of interviews with clients (site 1: 4, site 2: 1), parents/caregivers (site 1: 5, site 2: 4), and clinicians (site 1: 8, site 2: 9); minutes from clinician team meetings (site 1: 19, site 2 did not hold meetings), research team analysis meetings [22], parent advisor meetings [3], and researcher reflective notes across both sites. Descriptive statistics were generated regarding participant demographic details (clinicians, children and families), and percentage of numbers of clinician participants engaged in each data generation activity.

**Table 1. Overview and description of data sources used in the intervention.**

| Data source | Description | Quantity site 1 | Quantity site 2 |
|---|---|---|---|
| Observations | A trained ethnographer (author: **blinded for review**) sat in the room with the clients and clinicians during their clinical consultation. The main data source was the ethnographer's detailed observation notes. Notes detailed the minutia of clinic practices and processes such as: clinical procedures; clinic processes; interactions among/ between staff, clients and families; and the physical environment. | 36 | 8 |
| Dialogues | Transcribed audio recordings of dialogues. During dialogues, the researchers facilitated the clinicians through a process of critical reflexivity about their current practices and how they might be changed. | 10 | 5 |
| Interviews:<br>• clinicians<br>• clients<br>• families | Transcribed audio recordings of interviews | 8<br>4<br>5 | 9<br>1<br>4 |
| Family advocate consultancy | Meetings with a family advocate who is a mother of a child with MD attending one of the clinics. She gave input on study design, findings and reporting. | 3 (not specific to site) | |
| Team meetings | Researcher attendance at pre-existing 1-hr monthly clinician team meetings with their manager. | 19 | N/A |
| Analysis meetings | 1-2-monthly research team meetings to analyse incoming data across both sites. During these meetings the team iteratively conducted both inductive and theory-driven analyses of data | 22 (not specific to site) | |

## Analysis

Our analysis involved examining the 'outcome elements' we developed to consider the impact of the intervention. Namely:

1. **Clinician engagement in the intervention**

2. **Changes to how clinicians understood/characterised their roles and practices**

3. **Changes to clinician practices**

   We assessed each of these three outcome elements with the following sub-questions:

a. *For whom does the intervention work/not work and why*?

b. *What matters about how the intervention is carried out by the researchers in order for it to work*?

c. *What matters about the context (location/environment) into which the intervention is introduced in order for it to work*?

   The analysis included a number of iterative steps: Step 1–provisional analysis of all data sources in relation to the evaluation outcome elements (JS, BG and BM). Step 2–discussion and advance of the preliminary analysis with other research team members (TA, LM, DM and PT). Step 3—formal coding (using NVivo) of the empirical data by DM guided by the outcome elements. Step 4 –production of a detailed draft summary (JS) which was subsequently reviewed by the research team, participating clinicians, and an external advisory committee comprised of external clinicians, external disability and medical education scholars and parents. Insights from all sources were incorporated into the final analysis, which was discussed and agreed upon by the entire research team.

## Results

All clinicians at site 1 (n = 21) and site 2 (n = 20) agreed to participate in the study. Participants were from a variety of clinical disciplines at each site, with considerably more physicians from various specialities (n = 8: 40% clinicians) at site 2, in comparison to only two physicians (9.5% of clinicians) at site 1. Most clinicians identified as female (n = 36: 88%) across both sites. See Table 2 for demographics.

   The following discusses the results of our analyses across the three outcome elements (clinician engagement in the intervention, changes to how clinicians understood/characterised their roles and practices, changes to clinician practices), in response to the three evaluative sub-questions (*For whom does the intervention work*? *What matters about how the intervention is carried out in order for it to work*? and *What matters about the context into which the intervention is introduced in order for it to work*?). Results are summarised in Table 3, and explored in further depth below. Each outcome element is examined separately with the same subheadings used for each. Where relevant, results are supported by quotes/excerpts from the data using pseudonyms or generic terms (e.g., 'a clinician') for confidentiality.

### 1. Outcome element 1: Clinician engagement in the intervention

**1.1 Evaluation question A. For whom does the intervention work/not work and why?.** Although all clinicians participated in the clinical observations of their practices, participation in the dialogues (the key intervention strategy) varied both within each site and between the two sites. Attending dialogues involved a 2-hour meeting, roughly quarterly. A mean of 62% (n = 9) of clinicians (range 43–77%) attended dialogues at site 1, and a mean of 43% (n = 9)

**Table 2. Clinician demographic table.**

| Characteristic | Site 1 (n = 21) | Site 2 (n = 20) |
|---|---|---|
| Female (n, %) | n = 21, 100% | n = 15, 75% |
| Years of practice total (mean) | 13.1 | 18.6 |
| Years in child health(mean) | 9.7 | 15.6 |
| Years in this clinic (mean) | 5.2 | 5.8 |
| Clinical discipline (n) | | |
| Administration | 0 | 1 |
| Occupational-Therapist | 2 | 3 |
| Physician* | 2 | 8 |
| Physical Therapist | 3 | 1 |
| Psychologist | 1 | 0 |
| Recreation- Therapist/Life-skills Coach | 1 | 0 |
| Registered- Dietician | 0 | 2 |
| Registered Nurse | 3 | 1 |
| Respiratory- Therapist | 1 | 3 |
| Social worker | 1 | 1 |

*Physician specialities include: paediatric neurologist, paediatric physiatrist, orthopaedic surgeon, respirologist, paediatrician

clinicians (range 35–50%) at site 2. From our analysis of clinician interviews, the two most important factors relating to dialogue attendance were scheduling feasibility and perceived value. The project funded participation of clinicians (except physicians, who already had

**Table 3. Summary of results from the analysis.**

| Evaluation Questions ><br><br><br><br>Elements indicating the intervention 'worked' (or not) | A. For whom does the intervention work/not work and why?<br>• What influenced whether people participated?<br>• What were the outcomes for the various people involved?<br>• What outcomes were expected/unexpected? | B. What matters about how the intervention is carried out in order for it to work (and why)?<br>What were the critical aspects of *the project* that influenced how the programme operated? E.g. implementation, staffing, organisational context | C. What matters about the context into which the intervention is introduced in order for it to work (and why)?<br>What were the critical features of *the site* culture, belief systems, population groups, history etc that influenced what happened? |
|---|---|---|---|
| | | *NB: focus here is on **the project*** | *NB: focus here is on **the site*** |
| **1. Clinician engagement in the intervention** | • All clinicians working at the clinics for the evaluation period signed up to the study.<br>• All clinicians attended at least one dialogue and a mean of 9 (62%: range 43–77%) attended each dialogue at site 1, and a mean of 9 (43%: range 35–50%) at site 2<br>• level of participation appeared to be related to anticipated issues such as:<br>  • convenience for clinicians<br>  • perceived value of the project<br>  • comfort with critical reflexivity process<br>  • participation was considerably lower amongst physicians<br>• some new staff resisted participation (unanticipated issue) | • participatory methodology key to clinical relevance of the intervention<br>• research team which included clinicians, people with lived experience of the condition, group facilitation skills, social theory knowledge<br>• length of study is sufficient for changes to be made<br>• gatekeeper engagement was key, working with the team leaders from both sites from the outset<br>• engaging physicians is important, but challenging, requires incentives for physicians to attend<br>• project funds to pay participants<br>• Promotion of safety and trust amongst clinician participants including dealing with staffing turnover | •Institutional values and alignment so that there was conceptual support for the project<br>  • Access to institutional resources<br>  • Clinical team (and individual) readiness for the intervention |

*(Continued)*

**Table 3.** (Continued)

| 2. changes to how clinicians understood or characterised their roles and practices | There were considerable changes in how clinicians understood their role and practices as a result of the intervention. For example, this included:<br>• reduced emphasis on the biomedical goals and increased attention the 'human' aspects of living with MD<br>• reconceptualisation of 'risk' beyond physical risks to also include psychological and social risks<br>• Prioritisation of the emotional dimensions of the clinic (eg potential negative effects of pervasive positivity, lack of attention to 'negative emotions' such as grief) | • Promoting safety and trust amongst clinicians<br>• participatory methodology facilitated clinical relevance of discussions<br>• research team including clinicians, people with lived experience of the condition, group facilitation skills, social theory knowledge<br>• dealing with staffing changes<br>• high levels of consultation<br>• securing organisational support<br>• cohesive team dynamics assisted implementation | • Institutional values and alignment so that there was conceptual support for the project<br>• Access to institutional resources<br>• Clinical team (and individual) readiness for the intervention |
|---|---|---|---|
| 3. changes to clinician practices | *What influenced participation*: Engagement of team, individual engagement, personal interest in proposed changes. It helped that people could engage as much or as little as they wished (no pre-determined or expected level of participation).<br>*Changes included*:<br>a) changes to team processes e.g.<br>• Rounds: content includes 'human' elements<br>• Enhancing team values/purpose<br>• Increased valuing and utilisation of clinician experts in human-focussed clinicians<br>b) changes by individual clinicians<br>• risk discussions<br>• shifts in biomedical assessment prioritisation<br>• improved clinic experience for clients (e.g. reduce repetition)<br>NB: there was varying degree of uptake of these, these changes were not evident with all clinicians<br>c) changes for clients/families<br>• greater flexibility of clinic scheduling<br>• shorter clinic visits<br>*Expected/unexpected outcomes*?<br>Little was pre-determined so nothing was expected. We did not initially expect to be involved with implementing changes within the clinic.<br>*Data sources*: research team meetings, dialogues, clinician interviews, clinician business meeting notes | • time to make changes (long duration project)<br>• ongoing contact with clinicians (dialogues, clinician in team, attending team business meetings incl. team manager)<br>• changes were relevant to the context<br>• driven by clinicians<br>• the observations of clinical practice driving the discussion<br>• Partnership with medical team lead and other clinicians to keep changes relevant<br>• diverse and theoretically strong research team helped keep solutions creative/critical and feasible/relevant | • As the project is responsive to the context it would likely be able to be modified and implemented to meet the needs and particular constraints and opportunities in any healthcare context.<br>• There are no predetermined expectations about what change might be, and any changes are developed by the clinicians.<br>• There is however a need for some degree of institutional support and alignment (as above). |

protected research time) during normal clinical hours to facilitate their attendance, and we consulted with both teams to find suitable times to run dialogues.

In particular, a lower percentage of physicians compared to other clinicians attended the dialogues. Apart from the two site lead physicians who were part of the research team and who attended almost all dialogues, there was little physician attendance. This issue was more noticeable at site 2 because physicians comprised a higher percentage of the clinical team. Only two of the eight site 2 physicians regularly participated in the dialogues and only the lead physician attended all dialogues. Interview accounts suggest that the lack of physician engagement was likely due to time and scheduling constraints, although a more complex rationale

was offered by a few of the participants, summed up in this exemplar quote from a physician participant:

> I think part of it might be the hesitation of having someone observe you and then having to discuss it . . .I always think that [being busy is just an excuse], because we are always busy . . . you prioritize something else. And it may be fair enough to do that kind of prioritization, but it's not just because you're too busy.

At site 1, where most clinicians consistently attended the dialogues, both the researchers and clinical team commented on the engagement of the team as a whole and how this facilitated changes to clinic-wide processes. Discussing her experience across the study, a clinician noted the value to the team in terms of examining their shared values and team processes:

> As a team, we've had the opportunity to sit down and talk about how we want to serve our neuromuscular population and what are our approaches to care. And because we all came about this together, it works.

At site 2, a smaller group attended the dialogues, and as a result the participants expressed that they did not feel as if they had the agreement from their team members, and therefore lacked authority and/or broader insights to make changes within the clinic.

Even amongst those who regularly attended dialogues, the level of active participation across the study varied. For example, some participants had greater participation in dialogue discussions, and/or took leadership in making changes to clinic practices. Potential reasons for the varied levels of participation are discussed in the subsequent sections below.

**1.2 Evaluation question B. What matters about how the intervention is carried out in order to encourage clinician engagement (and why)?.** This section focuses on the critical aspects of *the project* that influenced how the intervention operated in terms of clinician engagement. Key elements are discussed under subheadings below.

*Participatory methodology.* The inherently participatory nature of the intervention helped clinicians actively engage in the project and enhanced its contextual relevance. As well as engagement in the methods embedded in the study (dialogues, observations, interviews), clinicians also participated in developing various study outputs–for example academic and non-academic presentations [29, 30], academic papers [5, 20, 27, 31] including this paper, and developing a clinician resource [32]. According to participating clinicians, this aspect set the project apart from other research projects conducted in their workplaces, none of which had provided them with opportunities to partner on shaping the outcomes that could affect their practices. The inclusion of client perspectives (via the observations, client and family interviews, and engagement of a family advisor) also helped engage the clinicians. One clinician described it this way:

> I think that [researchers] who have not been in clinic–it has a totally different feel . . . you guys have really picked up on what it is that goes on, and what the feeling is of the families, and the boys. . . Otherwise you constantly feel like you're talking to people who don't really get it, you know? So now you guys are on the inside of the circle.

*Research team.* The make-up of the research team was extremely important to the effectiveness of engaging clinicians in the intervention. It was beneficial that we included diverse researchers who had experience with: 1) the research topic both as clinicians and researchers (to keep the research relevant for clinicians), 2) lived experience of muscular dystrophy (to

bring valuable insights into client/family experiences), 3) group facilitation (to sensitively deliver the intervention), and 4) social and methodological theory (to ensure the intervention was rigorous, challenging and innovative—and thus interesting for clinicians). For example, a clinician at site 1 said:

> We talked before about getting a chance to reflect together on the bigger picture issues. I'm glad that we have the opportunity to do that because otherwise the time just isn't there, or the structure isn't there, to allow us to do that.

These types of comments were common across the two sites, expressing that these types of discussion were rare and valuable. From a site 2 clinician:

> I find each time we come it's like a mini retreat that we don't necessarily do when we set up our retreats annually–we set them up as an information session, and we do some discussion of the clinic function. But we don't get into the meat and the potatoes like I'm finding we have here. So I found it helpful, definitely reflective. . .. I find there's value in even just having discussed it, naming the elephants in the room.

*Length of the study.* The study was 3 years at site 1 and 1.5 years at site 2. The key benefits of the longer duration in terms of participant engagement were the depth of participant engagement with the conceptual elements of the project (e.g. 'human' dimensions), and more time to implement and reflect on changes.

*Engagement of team medical leaders.* Partnership with the teams' medical leaders was key. Both medical leaders agreed with the proposed research aims and approaches, and provided key input on how to tailor the project to their site. They were also involved as participants in all relevant data generation elements at their site. The medical leader at site 1 was also involved in most monthly research team analysis meetings.

*Engaging physicians.* As mentioned above, we had difficulty across sites engaging physicians other than the team's medical lead. At site 2, this was much more noticeable given the eight physicians on the team (as opposed to two at site 2). This meant that the dialogues regularly engaged only about half of the clinical team at that site. As a result, at site 2 we shifted the focus somewhat from team changes towards individual changes. Both aspects were always part of the project, so this shift was not difficult to achieve. However, a mechanism for advancing desired changes for the whole team might also be beneficial, such as partnering with managers.

*Funding clinicians' participation.* Grant funds were allocated to reimburse the hospital for clinicians' time in the dialogues and interviews. This may have meant that the institutions were more likely to support the project.

*Promoting safety and trust amongst participants.* As this project aimed to examine the hidden assumptions and unintended effects of clinical practices, the study had the potential to feel critical of individuals' or team efforts. For this reason, it was important to achieve the trust required for meaningful questioning of practices within the dialogues [33]. The participatory nature of the project, wherein clinicians were tasked with producing their own recommendations for change was integral to promoting trust. Elsewhere [5], we discussed other measures we used to develop trust including: gradual and considered introduction of potentially challenging issues to the dialogue discussions; reassurance of clinicians' expertise and the value they bring to the discussion; and modelling non-judgemental dialogue. Adjustments were made as needed. For example, after a particularly challenging dialogue we followed up with individual clinicians, as noted by a participant:

There was that one session that people found difficult . . . People were upset!. . .And I appreciated how you handled it and how you talked to everybody afterwards. You followed up, and did listen, and were receptive and responsive.

It was important to develop and maintain relationships. For example, a clinician highlighted the importance of having an unobtrusive ethnographic observer:

BM is so gentle, and such a just a lovely being, you know? I don't even know the word to describe it, but she is so non-offensive. I think that if you felt like this was a person who was, you know, keeping score, as if you were going to get a report card at the end, or whatever, I never felt like that.

**1.3 Evaluation question C. What matters about the context into which the intervention is introduced in order to encourage clinician engagement (and why)?.** This question focuses on the local contexts and sets of relations into which the intervention is introduced, in this case the two outpatient teams embedded in particular research-intensive hospitals. As the intervention was implemented in two different environments, we will discuss some key similarities and differences that mattered for clinician engagement. These included how well the teams already aligned with the conceptual underpinnings of the project, access to institutional resources, and team readiness for change.

*Institutional values and alignment.* Alignment of the project's conceptual underpinnings (enhancing the human aspects of MD care) with each sites' values and strategic plans facilitated the engagement of clinicians. Both institutions actively encourage research that partners with clinicians and directly impacts clinical care. Congruence at an institutional level meant that the interventional changes aligned with strategic directions of the institutions and made it easier to engender support from senior management for the project, including the teams' direct line-managers. Generally, this meant that clinicians felt that the aims underpinning the intervention were supported institutionally, at least on a conceptual level.

*Access to institutional resources.* On a practical level, engaging clinicians in this kind of intervention requires a degree of institutional resource support. Obviously, there needed to be a basic agreement for the study to be conducted within the institutions. Other structural support was also required to ensure clinician engagement such as: clinician release time for research; and physical space/infrastructure to conduct the dialogues, interviews and advisory meetings. It was very beneficial that both sites were academic health science facilities with embedded research infrastructure to support clinically-located research. It was also important to continue to negotiate support as requirements and possibilities were identified throughout the project. For example, at site 1 it was beneficial to have additional time allotted to discuss the project progression within usual clinician team meetings that included their manager, who then helped facilitate implementing the intervention. Addressing the project's dynamic needs at site 2 were more challenging, largely due to the lack of co-location with the research team (they were in different cities), as well as the absence of regular team meetings. This was framed by one clinician who noted that, despite numerous attempts, regular team meetings had not been a feature of that clinic for many years. She said the implications of this were that:

It doesn't give people the sense of working collaboratively. . . . People are called to the table when there's an actual issue, and do come. But on a regular ongoing basis, I think it's hit and miss.

*Clinical team (and individual) readiness*. The intervention required an immediate environment and team supportive of self-examination and change. This aspect of the intervention required clinicians to feel safe to be vulnerable and take emotional risks. These factors require a number of elements. For example, having fairly secure and stable teams with an established culture of trust amongst clinicians, existing traditions of open dialogue, a level of interest in the intervention, and familiarity and/or comfort with the reflexivity process. As in the following interview quote, most (but not all) clinicians said they became more comfortable with the reflexivity work over time:

> I'm very comfortable with it now. I distinctly recall at the beginning I felt uncomfortable in that you're really, really having to self-reflect when you see your words written on a sheet of paper and you're seeing how you're doing your interactions. . . Professionally it's been fantastic because now I'm able to sit back and really look at what I'm doing. I allow myself to be more critical about my actions but not feel judged.

Although both teams were well established, and were mainly secure/stable, when there was a high turnover of staff at about one year into the three-year study at site 1, the team felt less comfortable with the intervention. At site 2, there was little staff turn-over and this issue did not arise. During the period of staff turnover at site 1, there was resistance to participation from some new staff members. We did not anticipate this and likely did not adequately prepare the new participants for inclusion in the intervention. In retrospect, we did not account for: a) how far the project had developed (particularly in building trust with the initial participants), and b) the vulnerability of the team during the staff turn-over. About that time, one of the clinicians said during an interview: "Our team has been through some changes, and we're not the most resilient. . .so I think we're at a place where it's not the easiest to hear criticisms all the time. . .because everyone is tired, and working so hard." The staff changeover affected the comfort with which clinicians could critically analyse their work and their shared capacity to trial change. When there was a lack of team security or stability, the intervention benefitted from our attempts to increase the 'team-building' aspects within dialogues and other researcher/clinician-participant interactions.

Having a team with a high level of cohesion and open, regular group communication was important to the engagement of the clinicians for similar reasons. It is possible that a less cohesive team is one of the reasons why the physicians at site 2 were less engaged in the project. Unlike the site 1 team, the site 2 team had only some of the staff co-located (in particular, many physicians were often engaged with different teams as well), no common physical space in which they could congregate during clinics, and infrequent team meetings.

## 2. Outcome element 2: Changes to how clinicians understood/characterised their roles and practices

This outcome element focusses on changes to the ways clinicians *understood and/or* or *characterised* their roles and practices (i.e. not changes in *practice*–that will be the subject of the final outcome element 3 in the next section). The complexity, breadth and depth of the clinicians' reconceptualisations of their work was evident in the recommendations to enhance the human aspects of living with MD they co-developed at each site (see S1 and S2 Appendices). As discussed in the introduction, change in conceptualisation is often a precursor to enduring changes in practice.

**2.1 Evaluation question A. How does the intervention work/not work to change how clinicians understood/characterised their roles and practices?.**    There were numerous changes

in clinicians' understandings, and conceptualisations, of their practice. We describe three examples of key changes below.

*Increasing attention to the 'human' aspects of living with MD*. Clinicians demonstrated an increased awareness of their previous *focus on biomedical goals* and sidelining of the 'human' aspects of living with MD. For example, they noted that routine clinical processes were often designed to meet biomedical needs–such as timing clinics around medication reviews, nurses' checklists of disease progression markers; checks of adherence to prescribed home stretching exercises, diet, transfers, breathing exercises; and measurements of weight, height, muscle length, strength and physical function [5]. During dialogues, the clinicians discussed that the attention given to these biomedical goals might, at times, hamper working towards addressing the human aspects of living with MD. This change of thinking was evident in clinicians' discussions regarding how to shift the focus of clinic appointments. For example, at site 2 there was a discussion in dialogue 4 about how to increase attention to the cultural values of clients and families. One clinician said: "we need to have a better understanding of how can we walk this journey with you in your belief system without being neglectful or without harming the child". Similarly, at site 1 during dialogue 7, one of the clinicians suggested how the team might change the focus of pre-clinic rounds to extend beyond the traditional focus on nursing and medicine: "I think discussing what is going to be best for the clients is getting a whole history. We should be more inclusive with [other professions] speaking more. A good open discussion." Another clinician added to this discussion by proposing that, due to the time constraints during rounds, they might need to meet again to discuss some families when "we need to have a fuller discussion about a particular client". These suggestions by the clinicians show that they were thinking about restructuring clinic practices to better attend to the 'human' aspects of the lives of the patients and families without sacrificing medical needs.

*Broadening understandings of risk*. A reconceptualisation of risk was a key shift in the way clinicians thought about their priorities for client care. In early dialogues across both sites, discussions about risk were focussed almost entirely around minimising physical risks such as falls, fractures, reduced breathing function, or rapid disease progression. Clinicians articulated the need to 'push safety', even at the expense of rapport or provoking child and family guilt [20]. These earlier discussions rarely considered possible emotional or social risks. However, as the intervention progressed, a shift in thinking emerged and was evident in the clinicians' discussion in dialogue 6 (site 1):

> It comes back to that quality of life question that we asked way back at the beginning of this whole research project: What is quality of life for this child and this family? Is walking as long as possible and fitting in with your peers more important than the risk that they might fall?

This example was typical of the more complex discussions about various types of risk that clinicians discussed as important to consider with families.

*Prioritising emotion*. Another shift in the way clinicians conceptualised their work was a greater attention to the emotional environment of the clinic. This aspect of care was something clinicians said they had not considered much prior to the study. Clinicians discussed that they tended to focus on positive emotions, at times to the neglect of other emotions such as grief or anger [27]. They began to question whether being overly positive was always beneficial. One of the clinicians at site 1 said during dialogue 7 that her shift in understanding encompassed:

> the awareness of not being a cheerleader all the time or turning everything positive. . . . If somebody says something about a grief or a loss or something, rather than trying to put a positive spin on it, to acknowledge and validate [it].

Clinicians increasingly spoke of a greater attention to supporting clients and families through a range of emotions.

Participating clinicians suggested fostering a 'positive environment' may have meant they gave less attention to discussing 'difficult topics' with their clients and families, for example, disability stigma, clients' sexuality, or decline and death [25]. At site 1, it was evident that some of these conversations were re-prioritised. For example, in dialogue 8, a clinician said: "we're trying to create . . .a healthy culture of talking about death and decline". In this way and others, there were greater considerations of the emotional environment of the clinic [26] and its implications for clients and family.

At site 2, participating clinicians suggested that they already attended to a full range of emotions prior to the commencement of the study, though that was not coordinated but the practice patterns of particular professionals. Pro-active discussions of 'difficult' topics were evident in the observations from early in the study:

> The physician then asked Casey (aged 8) if anyone ever "bugged or teased" him. Casey sort of nodded and looked up at him. The physician looked at Casey sympathetically but in a supportive way. Casey said sadly, "I'm not very good at tag." The physician did not say anything as Casey looked back at the stethoscope. The physician looked up at mom and then back down at Casey and said in a supportive way, "School should be a safe space," and asked if anything happened such as bullying.

Different approaches to tackling difficult topics with clients is an example of a benefit of our two-site design: it made possible the sharing of issues, solutions, and (de-identified) examples across the teams.

**2.2 Evaluation question B. What matters about how the intervention is carried out in order to change how clinicians understood/characterised their roles and practices?.** Our analysis of this question aligns with 1.2 above. Elements such as repetition, the mixed make-up of the team, high levels of consultation, and time and organisational support were important to the impact on clinicians' understandings. Furthermore, aspects such as cohesive team dynamics assisted implementation; as did early adopters who encouraged changes in the clinic and with their colleagues, and inclusion of team medical leader in research meetings.

**2.3 Evaluation question C. What matters about the context into which the intervention is introduced in order for it to change how clinicians understood/ characterised their roles and practices?.** As per the response to 1.3 above, institutional conceptual support for the project, access to institutional resources, and clinical team (and individual) readiness for the intervention were important elements to create possibilities for change in clinician conceptualisation.

## 3. Outcome element 3: Changes to clinician practices

**3.1 Evaluation question A. How did the intervention work/not work to change clinician practices (and why)?.** There were numerous practice changes made by clinicians and the clinical teams. It was vital that participants had sufficient autonomy to make changes to their individual practices, and team operations. At both clinics, clinicians largely had autonomy to change their own practices. However, a more intensive and longer engagement of the site 1 team helped them to institute individual and team changes. The site 2 team discussed more difficulties approaching clinic-wide changes, in part because of reduced team participation in dialogues and lack of regular meetings to follow-up on team implementation. However, it was helpful that there was no pre-determined or expected level of participation. Flexibility meant

that busy clinicians with multiple priorities could engage as much or as little as they wished, or as time allowed, ensuring that we could meet people 'where they were at'.

Below we provide examples of the key changes in clinician practices under the following headings: a) *changes to team processes*, and b) *changes to individual practices*. We then discuss the c) *impact on clients*.

*a) Changes to team processes.* There were changes to team processes at both sites, but more at site 1 likely due to the team's greater and longer engagement. For example, at site 1, there was an overhaul of how clinical team meetings, commonly known as 'rounds', were organized. Extra rounds were added to ensure that all clients were attended to, and there was a shift in focus from primarily discussing medical information (diagnosis, medications) towards greater consideration of the 'human' aspects of living with MD. Site 2 had irregular rounds. This was highlighted in the dialogues as something that they would like to change and were actively working towards implementing. At time of writing, constraints on physical space, and scheduling rounds to accommodate physician team members persist at site 2; regular rounds have not yet been instituted.

There were also notable changes to enhance team purpose at site 1. By their own volition, they developed a 'mission statement' to encapsulate inclusion of the 'human' aspects of living with disability into the clinic. They felt this would consolidate a shared approach and set of values that would encourage and support the team to inform their practices in the long term. Their mission includes a commitment to "respectfully support and guide children/youth and families to reach their personalized life goals and values." This re-purposing is evident in a number of changes to the clinic; for example, one of the clinic nurses now calls families before clinics to ask what they want from their upcoming appointment. This practice, which was already in place at site 2, facilitated a focus on child and family priorities in the visit.

Finally, there was an enhanced recognition and utilisation of team clinicians whose roles and expertise focussed on non-biomedical aspects of living with MD, e.g. social work, psychology, and recreation therapy. At site 2, the key change was a shift to allocating fixed times for clients and families to see a range of clinicians. Traditionally, only physicians were given fixed appointment times. This change resulted in more time being devoted to non-medical issues of clients and families. At site 1, this process was already in place but in the dialogues, the team discussed how they had shifted their practices to support the work of the recreational therapist:

> Clinician 1: I've been working more with [therapist] and trying to set up the environment for people to be open to coaching about things that are important to them, be it friendship, be it transition. [. . .]

> Recreational Therapist: It's worked well, because it used to be if I was going [to see a family] after [Clinician 1] I felt like they had their bags packed ready to go [i.e. leave the clinic]. And now when we were having this conversation and talking about it the importance of what I can do through gaining [life/social/occupational] skills.

The site 1 clinical team also began to engage the social worker in early appointments in recognition of the value of her training in providing emotional support while families are adjusting to the new diagnosis for their child.

*b) Changes by individual clinicians.* There were numerous changes evident within individual clinician's practices which often related to new understandings or characterisations of their practices. Below we elaborate on the changes we discussed in section 2.1: risk discussions were more nuanced; routine biomedical assessments were not automatically prioritized; and there was decreased repetition within appointments.

Discussions about risk became more nuanced in the clinic observations over the course of the study at site 1, which reflected the nature of the conversations in the dialogues. The clinicians often moved beyond a focus on physical risks to also consider how risks might also be social or psychological. For example, in one appointment, two clinicians discussed with the parents of 'Billy' (aged 4 years) the possibilities of restarting corticosteroids (a medication used to slow the progression of MD symptoms and thus extend lifespan). The family had tried the medication but had decided to wean off the medication because of its considerable negative side effects that made Billy hyperactive, restless and unhappy. It was a difficult decision for the family with the mother expressing that she 'wanted her son back'. During observations, our research assistant noted:

> Mom described how Billy would keep everyone up all night and said they just can't handle going back to that. The clinicians both agreed with the family—they all needed to sleep. . .One clinician then added that they should "do what is best" for the family and that she would leave it with the family to start the steroid if they wished . . ..Later in the conversation, she again reassured the family by explicitly stating she was "okay with them starting or not starting" the medication, and saying that they needed to "balance everything" in this decision.

Although there are physical benefits to taking corticosteroids, the clinicians strongly demonstrated non-judgemental support for the decisions and priorities of the family to enhance the interactional and social aspects of their life in this interaction. They clearly demonstrated to the family that they would not negatively judge them if they continued to prioritise their family and child's psycho-emotional wellbeing over slowing physical progression of MD and longevity.

This shift to attending to the human aspects of living with MD was also evident in observed discussions with families about which activities/play children should or should not engage in. Clinicians were more likely to not only talk about physical risks, but also the psychological and social aspects of engaging (or not engaging) in these activities.

Although biomedical assessments remained a primary focus of clinic appointments, there was a noticeable shift towards attending to the human aspects of living with MD. This included setting aside routine biomedical assessments and increasing time and attention to issues of psychological and social wellbeing when relevant. For example, an observed clinical consultation with Hayden (age 9 years) and his dad started with a clinician asking how Hayden was doing. His dad responding that everything was 'emotionally well' with the family. The clinician responded by saying that if Hayden wanted to, he could talk about anything he wanted to, either with her or the social worker. Perhaps due to this invitation, Hayden's dad later raised that there had been an unexpected death of one of Hayden's grandparents. At this, the clinician again made it clear that this conversation was welcome in the clinic, by saying: "Let us know how we can help your son or you as a family. It's important." The observer recorded the following exchange:

> Dad said, "that's why I brought it up." And then he began to explain how each member of their family was dealing with it. He said that his wife was obviously upset but then flew into 'work mode' to help get things sorted out; his daughter cried and talked about it. Hayden had not shown much emotion though did say he understood what happened and he was sad. They had been watching both of the kids since they found out, and are a little worried about Hayden.

These kinds of conversations were more evident in the observations over time at site 1: clinicians invited clients and families to discuss human elements of living with MD.

The increasing attention to making the site 1 clinic less uncomfortable for children and families was evident in both the clinic observations and the interviews with the clinicians (near the end of the project). These efforts recognised the complexities of living with a child with MD–particularly those living far from the clinic. The two to four-hour clinic visits can be very tiring. For example, a clinician said that, as a result of the dialogues, there has been "more of an effort to . . . decide which clinician needs to see which client . . . trying to make the clinic appointment as short as possible." Other approaches included reducing the repetition between clinicians, something site 2 also planned to pursue. These changes sought to addresses problems of child anxiety around clinic visits and a frequent lack of child engagement identified in the observations and interviews.

*c) Impact on clients (young people with MD and their families).* The ultimate aim of the intervention was to improve the lives of the young people and their families who access these types of services. We did see evidence of how the changes benefitted clients. For example, reduction of clinic length may seem like a small change to practice, but the accounts suggest it made a difference to children and families' experiences of clinic. A site 1 clinician stated in her interview:

> One of the really tangible outcomes [of this study] is we are a little more conscious of time. I've had numerous families [say that they] appreciate that the clinic visits have been shortened or there's been more cohesion in transfer of information, less repetition.

The impact of changes to clinic processes and logistics were also evident in the observations. For example, in an observation of Kyle (age 11) and his family, the clinicians provided an option to conduct part of the appointment by phone. The observer recorded Kyle's parents' responses to this option:

> Mom said "That's a really great idea." She said doing it in that way would be easier for them to process what is being said [as clinic times were so busy] . . . Dad also said that would be a much better way to process everything.

In a subsequent observed appointment, a few months later, Kyle's mom said that shorter and less repetitive appointments had helped to make the clinic visit more manageable:

> Mom said she really liked that they had streamlined it for them. Her accompanying friend even commented that it was much faster than other times she had come. She noticed that they had not repeated things this time which frustrated her every time she came with the family. Mom said that one of the nurses had called twice since their last appointment and did a lot of the nursing stuff over the phone such as Kyle's medication. She then told the nurse exactly what they wanted to discuss at this appointment. The nurse also talked to mom about the things the team wanted to talk about. She told the nurse that she did not want any residents in the room. She understood that they were learning . . . but found that it was very tedious to go through things with them. It's also a long drive for them so she said they needed to get in and out as quick as they can. She said they like this process much more than before.

The clinicians' flexibility to conduct appointments differently resulted in the family providing more input into the clinic visit, crafting a visit that suited them better.

**3.2 Evaluation question B. What matters about how the intervention is carried out in order for it to change clinician practices (and why)?.** The duration of the project was important. Changes to clinical practices, particularly those that require significant shifts in thinking and practices, take a long time to institute. Thus, a longer duration project had more impact. At site 1, more changes were possible with the 3-year duration. As mentioned, co-location was also key. Site 2 had a 1.5-year timeframe (due to funding) and less was possible. The longer duration at site 1 also allowed more opportunities to partner with clinicians and together identify, try out, and consolidate shifts in practice. This happened during dialogues and observations, by engaging clinicians in the research processes, and by attending monthly team business meetings (site 1).

It was also important that changes were relevant to the context, were feasible, and were of interest to the teams. Two key factors drove the relevance and feasibility; 1) having the participatory input from clinicians, including their medical leaders, and 2) drawing on clinical observations to drive the discussions and co-analysis. As discussed earlier, having a diverse and theoretically strong research team with facilitation skills helped keep solutions creative/critical and feasible/relevant.

**3.3 Evaluation question C. What matters about the context into which the intervention is introduced in order for it to change clinician practices?.** Thus far, this evaluation has addressed the contextual particularities, how they shaped the intervention, and their impact throughout the analysis above; to avoid repetition they will not be reiterated here. Key elements were the flexible and responsive design of the project that ensured that the approach could be modified and implemented to meet the needs and particular constraints/opportunities of these healthcare contexts. The participatory approach ensured that any changes were developed by the clinicians themselves. The willingness of the clinicians to participate, and the institutional support and alignment, also promoted possibilities for changes in clinician practices.

## Discussion

Our evaluation suggests that the intervention was largely successful in engaging clinicians in the human aspects of living with MD and the applications to clinical care. This work responds to the now considerable transdisciplinary research findings, including our own, that highlight a need for a shift in focus of healthcare on biologically repairing/adapting the body to the exclusion of the human aspects of care [e.g., 34–37]. The interventional approach could potentially be used by other researchers, clinicians or service designers to facilitate this shift, and related persistent issues such as physician-centred approaches and underutilisation of the other resources within multidisciplinary teams. The attention that both the research methodology and evaluative approach provide to the specifics of the context is likely to be vital to creating a shift in such imbedded practices. This includes the flexible, participatory design and evaluation; diverse makeup of the investigator team (e.g., clinicians, consumers and social theorists); longitudinal timeframe; and use of applied critical reflexivity as both a study approach and teachable skill.

As further evidence of the impact of the intervention on sustainably shifting the focus of rehabilitation practices beyond biomedicine, there have been some clear impacts beyond the immediate context of the project. For example, the large research centre at one of the sites has included the human dimensions of living with disability as one of its three primary themes. Further, one of the site hospitals explicitly drew from the study results to inform a redevelopment of all of its outpatient services.

There were a number of challenges to delivering this intervention that provide lessons learned for adapting it to other contexts and teams. First, we did not anticipate that we would

be conducting a multi-year interventional study. Our original project focused on co-identifying what practices were amenable to change and formulate recommendations for local (site 1) and similar MD clinics. However, the participating clinicians were keen to move the study thinking beyond the conceptual stage, so we adapted the study, and secured additional funding, to create a longitudinal intervention and evaluation. This was a welcomed progression but created some challenges in advancing the work that we highlight below. A second challenge was time. We were only able to secure funding to extend the project at one site. Thus, for similar interventions, we would recommend that implementation plans include several months of ongoing facilitation and supports. In our case this was three years. A third challenge, but also a strength of the design, was remaining flexible in how changes were implemented. Finally, a key challenge was the engagement of physicians. We would suggest addressing the participation of any busy and/or disengaged clinicians within the project design, perhaps by using incentives, discussing it with them privately, or conducting the study contexts where physicians/clinicians already regularly attend meetings. However, the distributed emphasis of the intervention on the human aspects of care may support shifts in clinic practices even if more biomedically focussed clinicians are not involved.

When considering the implications of this study, it is important to be aware of the conceptual basis of a realist evaluation, our theoretical underpinnings and the claims they support. This type of project and evaluation does not attempt to find a stable or singular truth, but to enhance understandings of how and why an intervention might work (or not work). The impact of the intervention needs to be considered as situated in the context in which it was delivered. This means that the same results may not be produced elsewhere using the same procedures [29], and that a flexible approach is the key to success. It is also important to consider how the authors (who each have their own habitual thinking and practices) influenced the reporting of the results of the evaluation. We are a mixed bunch, including physiotherapists, sociologists, physicians and students. One of us lives with MD. Although a number of us are clinicians, we come from different positions and academic commitments that meant we designed the intervention and focused our analyses and interpretations of success/impact in particular ways. Our diversity also means that we produced a range of outputs regarding how to interpret and apply our shared learning for different audiences. This includes conceptual writing on clinical objectification practices [28], emotions [26, 27], living with death/dying [25], clinical papers regarding shifting practices [27, 31], a methodological study on fostering reflexivity with clinical communities [20], and clinical ethics [28]. We also created a non-academic resource to support clinicians' critical reflexivity [38]. The wide variety of outputs that draw on our different interpretive frames and disciplines is a strength of this project.

There are innumerable possibilities opened by this type of intervention. Our intervention process is innovative, including its level of ongoing participatory engagement, its flexibility, and the involvement of critical reflexivity and social theory. The intervention works to *shift thinking first*, *and practices second*, with improving clients' lives as the ultimate end point. Due to these qualities, the intervention works to produce creative solutions that challenge assumptions underpinning clinical practice. As the process involves investigating what is going on in any given context, the intervention could feasibly be applied in any setting.

## Supporting information

**S1 Appendix. Abbreviated list of recommendations (site 1).**
(DOCX)

**S2 Appendix. Abbreviated list of recommendations (site 2).**
(DOCX)

## Acknowledgments

The authors wish to thank the study participants and advisors.

## Author Contributions

**Conceptualization:** Jenny Setchell, Laura McAdam, Patricia Thille, Thomas Abrams, Bhavnita Mistry, Barbara E. Gibson.

**Data curation:** Jenny Setchell, Donya Mosleh, Laura McAdam, Patricia Thille, Thomas Abrams, Bhavnita Mistry, Barbara E. Gibson.

**Formal analysis:** Jenny Setchell, Donya Mosleh, Laura McAdam, Patricia Thille, Thomas Abrams, Hugh J. McMillan, Bhavnita Mistry, Barbara E. Gibson.

**Funding acquisition:** Jenny Setchell, Laura McAdam, Patricia Thille, Thomas Abrams, Hugh J. McMillan, Bhavnita Mistry, Barbara E. Gibson.

**Investigation:** Jenny Setchell, Donya Mosleh, Laura McAdam, Patricia Thille, Thomas Abrams, Hugh J. McMillan, Barbara E. Gibson.

**Methodology:** Jenny Setchell, Laura McAdam, Patricia Thille, Hugh J. McMillan, Bhavnita Mistry, Barbara E. Gibson.

**Project administration:** Jenny Setchell, Donya Mosleh, Laura McAdam, Patricia Thille, Hugh J. McMillan, Bhavnita Mistry, Barbara E. Gibson.

**Resources:** Laura McAdam, Hugh J. McMillan, Barbara E. Gibson.

**Software:** Barbara E. Gibson.

**Supervision:** Barbara E. Gibson.

**Writing – original draft:** Jenny Setchell, Bhavnita Mistry, Barbara E. Gibson.

**Writing – review & editing:** Jenny Setchell, Donya Mosleh, Laura McAdam, Patricia Thille, Thomas Abrams, Hugh J. McMillan, Bhavnita Mistry, Barbara E. Gibson.

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
