## [Decision Letter · Decision Letter 0]

24 Feb 2021

PONE-D-20-30857

Enhancing human aspects of care with young people with muscular dystrophy: A realist evaluation of a participatory qualitative study with clinicians.

PLOS ONE

Dear Dr. Setchell,

Thank you for submitting your manuscript to PLOS ONE. After careful consideration, we feel that it has merit but does not fully meet PLOS ONE’s publication criteria as it currently stands. Therefore, we invite you to submit a revised version of the manuscript that addresses the points raised during the review process.

Both reviewers made useful general and specifc comments which I will like the author team to adress.

In addition, as an Editor I feel it will be usefll for both reviwers if you can unbind the previous studies with the full reports on the methodologies so that reviewers  will have a thorough appraisal of the study’s methods. Also provide more detail explanation of the ‘human’ aspects of care under the ‘introduction’ section of the study to further highlight the relevance of these aspects in the management of MDs.  In as much I do not want to the authors to write a lay paper I will agree with reviewer 2 to edit/simplify the paper in places so that complex information and messages comes across to readers in a clear and understandable way. I will also want to see clarifications about the methodology from an RE perspective and explicit use of the terminologies and language of RE or a clearification that the study was not a true RE. Can the authirs address while initial theory was not laid out. Perhaps some of the questions can be answered if reviewers have access to the method  proper.

We look forward to receiving your revised manuscript.

Kind regards,

Ukachukwu Okoroafor Abaraogu, BMR PT, MSc, PhD

Academic Editor

PLOS ONE

Journal Requirements:

3. Thank you for including your ethics statement: "The human research ethic boards from both study sites approved the study. Holland Bloorview: 17-729 CHEO: 17-117X Written consent was obtained from all participants."

4. Please ensure you include additional information regarding the survey or questionnaire used in the study and ensure that you have provided sufficient details that others could replicate the analyses. For instance, if you developed a questionnaire as part of this study and it is not under a copyright more restrictive than CC-BY, please include a copy, in both the original language and English, as Supporting Information, or include a citation if it has been published previously.

Reviewers' comments:

Reviewer's Responses to Questions

**Comments to the Author**

1. Is the manuscript technically sound, and do the data support the conclusions?

Reviewer #1: Yes

Reviewer #2: Yes

2. Has the statistical analysis been performed appropriately and rigorously? 

Reviewer #1: Yes

Reviewer #2: N/A

3. Have the authors made all data underlying the findings in their manuscript fully available?

Reviewer #1: Yes

Reviewer #2: Yes

4. Is the manuscript presented in an intelligible fashion and written in standard English?

Reviewer #1: Yes

Reviewer #2: Yes

5. Review Comments to the Author

Reviewer #1: The study evaluated a participatory interventional procedure geared towards enhancing ‘human’ aspects of MD management among clinicians. The study is relevant to literature and seems to have sound methodologies. However, the blinding of the previous studies with the full reports on the methodologies limits a thorough appraisal of the study’s materials and methods. Shallow explanation of these ‘human’ aspects of care under the ‘introduction’ section of the study failed to highlight the relevance of these aspects in the management of MDs. Below are specific comments.

Title: This is apt for the study.

Abstract

1. Purpose: Is it entirely correct to call this study a ‘3-year’ study considering the fact that the study lasted 1.5 years in one of the two centres?

2. Purpose: The last sentence under this section looks more like a ‘methods’ statement rather than a ‘purpose’ statement.

3. Results: As a follow-up, can the authors list ‘why’ there were variations in the ways clinicians changed their thinking and practices?

Introduction:

1. The first statement under the introduction was the aim of the study. This is uncommon and unfamiliar in the scientific world. This may be more apt as the concluding statement of the last paragraph under introduction, which the authors had already done.

2. Assessing and improving the ‘human’ aspects of care is the focus of this study. Despite this, authors failed to elaborate on these aspects of care under the introduction. Authors may also need to highlight that these parameters are common and are usually neglected in MD management. This will help to better justify the study.

Materials and Methods

1. Authors reported that the methodologies of the present study had been reported elsewhere (references number 5 and 14). However, these references were blinded thereby making it difficult for the reviewer to consult these articles. Can the authors please unblind the blinded references in order to facilitate better review process?

2. The Intervention: How many series of the 2-hour meetings were conducted in each case? What are the spacings (time lag) between two consecutive interventions? These questions were probably answered in the previous publications but there was no way the reviewer could ascertain that since the publications were blinded. Though these were mentioned under the results, they are supposed to be talked about under methodology.

3. Study sites and participants: The sentence ‘Most significantly, the study duration was considerably longer at site 1 versus site 2 (3 and 1.5 years respectively)’ seems confusing. Authors should please consider rephrasing the sentence.

4. With the reported difference in the duration of the study at the two sites, it is logical to think that some procedures were shelved for site 2. How did the procedures (especially the interventions) in the two sites differ? What precautions were taken to make sure that the differences in procedures did not affect the validity of the study?

Results

1. Detailed and reported in a coherent manner.

2. Authors cited table 3 before table 2 intext. Reverse should be the case. Alternatively, authors can redesignate the tables accordingly.

3. Table 2: There is a repetition in the cell where the second column intersects with the second row. Please correct: “……clinicians attended each dialogue at site 1, and a mean of and a mean of 9 clinicians…..”

4. I am just thinking if this quote on page 11 not better suited for the third sub-question (item 1.3) instead of the first:

It doesn't give people the sense of working collaboratively. … People are called to the table when there's an actual issue, and do come. But on a regular ongoing basis, I think it's hit and miss.

5. Page 24: The word ‘to’ before the phrase “do what is best” needs to be expunged from the quote. The affected aspect of the quote is shown below:

‘…..One clinician then added that they should to “do what is best” for the family and….’

Discussion: Well-written

Reviewer #2: FEEDBACK

TITLE: Enhancing Human Aspects of Care with Young People with Muscular Dystrophy: A Realist Evaluation of a Participatory Qualitative Study with Clinicians

OVERALL FEEDBACK:

Thank you very much for the opportunity to review this manuscript. I would like to congratulate the authors on an extremely insightful study that I think has wide applicability, well beyond children and youth with MD. I have tried to provide thoughtful and constructive feedback that I hope will improve the impact of the manuscript. I have two main points of general feedback, followed by specific feedback broken down by section.

GENERAL FEEDBACK:

1. The manuscript is very well written overall. However, at times it is a lot. There is NO DOUBT that this group of authors are extremely deep and complex thinkers. But an important skill is being able to take a lot of complex information and distill it in a clear and understandable way. I wonder if the authors could re-read and edit the paper with this in mind. When I was doing my PhD, my supervisor used to say, “be kind to the reader”. For example, I personally love Bourdieu and agree that his thinking and philosophical concepts can provide a relevant lens through which to view theory and practice in rehabilitation. However, given the density of the manuscript I think the authors can make the point about practice change in a way that is much more succinct and accessible, thereby paring down the intensity and length of the manuscript.

2. The authors report using a realist evaluation (RE). I do see RE imbued in the methodology, including the way that the research question is written. However, there are almost NO explicit talk about RE, no talk about context, mechanism, and outcomes, and context-mechanism-outcome configurations, nothing about initial program theories etc. etc. I see it more clearly in some respects but not in others. For example, typically RE begins by creating an initial program theory (drawing on program architects and/or realist syntheses) which consists of several C-M-O configurations. Then, the initial program theory is tested via the research, then refined in response to the research. The authors describe the four steps in their data analysis (i.e., Step #1-provisional analysis of all data, Step #2, discussion and advance of the preliminary analysis), but as the reader I am wondering “Does this mean they created an IPT?” If so, how, specifically? Despite the authors referring to RE and some of the characteristics of RE being evident to a trained eye, the methodology from an RE perspective is opaque. I wonder why the authors chose not to be explicit and use the language of RE. This approach could undermine the veracity of the claim that this was an RE study and therefore ultimately undermine the quality of the work. Although I do not doubt the quality, I suggest that if it is an RE, be explicit. If it was not a true RE then be explicit about that as well, refrain from calling it an RE, and decide on a more accurate term. The other associated issue is that I feel that RE has a huge potential to contribute to the development of knowledge in rehabilitation. But, if as authors and researchers we are opaque, how can we advance the use of this methodology? Part of the contribution of this paper are the findings related to MD, to be sure, but a big contribution could also be in demonstrating and providing an exemplar of the use of RE in rehabilitation.

INTRODUCTION:

3. The introduction would be strengthened if the authors made clear, from the opening sentence, what kind of intervention they are referring to. The first line states, “This paper evaluates an intervention” then throughout the paragraph the researchers talk about “health care” practice “clinical practice”, an “intervention in clinical sites” and after reading the whole introduction I, as the reader, still had no idea about what the intervention is and what type of health care professionals are involved. In addition, several of the references cited in the introduction relating to “clinical practice” broadly (i.e., #7, 13) actually relate to primary care and mostly physicians in primary care (as per references #7 and #13). It is recommended that the authors clarify what type of intervention they are referring to from the very beginning AND include references, where possible, from the body of literature of all the professionals involved that support to the content of their introduction, not just primary care (physicians).

4. As stated in #1 above, the paragraph from “Clinical practices” ending in “contextually relevant” could be very easily simplified and shortened by tightening the writing.

5. For the sentence in the introduction that states “Even when there are clear parameters…” and ends in “…in part because what is required to implement changes in one context varies from the next [8]. I understand why, given that the authors are building towards rationalizing the methodology (RE), the focus on context. However, the sentence feels unbalanced given that there have been many documented reasons (across health care professions) why practice change is difficult. So, to be fair-minded and balanced, I suggest that the authors indicate, even in parentheses, that there are many documented reasons why practice change is difficult (e.g., ) and after the change is made in #1 above, use citations that are relevant to the fields of practice under study. THEN, comment on differing contexts being one such important factor. Also, given that the study is a RE, consider including a reference from the RE literature to support this point. As stated above this whole paragraph can be simplified while still making the points that (a) practice change is hard to shift for many reasons, (b) one of the main reasons relates to the varying contexts within which programs are designed then subsequently delivered.

MATERIALS and METHODS:

6. The authors refer to “dialogical exchange”. When I read “dialogical” I immediately think “Bhaktin”. Then, there is no description about what is meant by a “dialogical exchange” or reference provided. What are the specific criteria or parameters that made it a “dialogical exchange” versus a discussion or interview? Please be clear, perhaps adding a reference, b/c I am left wondering what the guiding lens was for the interaction between the researchers and the clinicians at each site. I apologize if this information is reported elsewhere in other publications detailing the project methodology. The references have been blinded so I cannot cross-reference. In any case, I think it is worth clarifying here, at least with a reference, to make it more understandable to the reader, particularly the reader from a non-academic audience.

This concern came up again under the section titled “The Intervention”. It seems the dialogues are part interview, part mentorship and coaching. Please add a reference for what was done or an explanation of how the “dialogical protocol” was determined and how it was anticipated that this particular approach would be suitable to elicit practice change. Methodologically, it is not clear what the process for determining the “dialogical exchanges” was based on, particularly since there is not one reference provided regarding dialogical exchanges. It would help if the authors, even in one brief sentence, made it clear how they came up with what would be the parameters of the “dialogical exchange” or at the very least provided a reference. It would also be interesting and important to know the specific qualifications of the researchers to be providing this type of “critically reflective mentorship” (I will call it for lack of a better term) to the clinicians. The authors run the risk of coming across as somewhat paternalistic in their role as researchers to the clinicians unless they can explicate some specific set of skills that the particular researcher possesses or some specific resources that they drew on OR how they addressed this issue. I do not have access to reference #14 b/c its blinded but maybe include at least a comment that measures to cultivate trust were implemented, however should also be made explicit here.

Although the authors do address this issue in the RESULTS section under 1.2 Evaluation Question B, I think it is worthwhile adding one statement that addresses this issue in the Methods.

METHODOLOGY and THEORETICAL UNDERPINNINGS:

7. The first sentence of this section states “In alignment with the critical social science underpinnings of our study, we used a realist approach to evaluate the intervention”. I think that sentence should be deleted. The philosophical underpinning of realist evaluation is scientific realism not critical realism. The authors may be referring to Bhaskar’s philosophy, but Pawson differentiated realist evaluation (and scientific realism) from Bhaskar and critical realism. The authors can refer to Pawson (2006) Evidence-based Policy: A Realist Perspective for more information, or just delete the first sentence Furthermore, ontological, and epistemological congruency with the larger study are not a prerequisite for RE.

RESULTS:

8. The authors did an EXCELLENT job of reporting qualitative findings - describing themes that are conceptually deep and sufficiently abstracted up from the data, then using well suited quotes judiciously to illustrate the findings. The findings are absolutely fascinating, so insightful, and so important. This section was a pleasure to read!

9. I could be completely missing something BUT in the manuscript under results it states, “All clinicians at site 1 (n=21) and site 2 (n=20) agreed to participate….See Table 3 for demographics”. Then, when I go to Table 3 under Site 2 it says n=15, 75%.

10. Question 1.1 “For whom does the intervention work” is about outcomes. Then later in that section there is a sentence about the level of active participation was continent upon (1) clinician’s level of interest, and (2) familiarity with reflexivity. To me, based on definitions from RE, these are pre-existing characteristics of the actors or agents and should therefore be considered as part of the context required for the program to elicit outcomes.

11. A minor point but I think there is a word missing under section 3.1 (a) Changes to team process. There is a sentence that I believe should read “Extra rounds e ensure that all clients who were attended to were added…”

DISCUSSION:

12. Lower physician engagement was referred to several times within manuscript, and the intervention seemed to potentially shift practice from a bio-medical, physician centred approach to stronger support, recognition, and utilization of the strengths of all team clinicians, primarily in addressing the human aspects of living with MD. This is really important b/c issues with physician engagement, the prioritization of physician-centred/bio-medical approaches, and the emphasis on the physician voice despite the resources of multi-disciplinary teams has been a long-standing issue in rehabilitation practice. I was very excited to read about the shift and therefore looking for the authors to comment on how the intervention can be used to address the longstanding issue and how RE contributed to understanding this important shift in a way that can be implemented by others. The authors talk about these shift in relation to their study/context, but I was looking to see how the authors thought their findings could inform advancement of knowledge regarding these persistent issues.

6. PLOS authors have the option to publish the peer review history of their article (what does this mean?). If published, this will include your full peer review and any attached files.

Reviewer #1: No

Reviewer #2: No

---

## [Author Response · Author response to Decision Letter 0]

20 Jul 2021

Many thanks to the reviewers for their comments on our manuscript. Please see the attachment for our detailed responses.

---

## [Decision Letter · Decision Letter 1]

2 Feb 2022

Enhancing human aspects of care with young people with muscular dystrophy: An evaluation of a participatory qualitative study with clinicians.

PONE-D-20-30857R1

Dear Dr. Setchell,

We’re pleased to inform you that your manuscript has been judged scientifically suitable for publication and will be formally accepted for publication once it meets all outstanding technical requirements.

Kind regards,

Ukachukwu Okoroafor Abaraogu, BMR(PT), MSc, PhD

Academic Editor

PLOS ONE

Additional Editor Comments (optional):

I am happy to conditionally accept this manuscript pending corrections in line with the minor comments of the reviewers. The manuscript does not need to go another round of external review but I will like to see the corrections before it proceed to publishing 

Reviewers' comments:

Reviewer's Responses to Questions

**Comments to the Author**

1. If the authors have adequately addressed your comments raised in a previous round of review and you feel that this manuscript is now acceptable for publication, you may indicate that here to bypass the “Comments to the Author” section, enter your conflict of interest statement in the “Confidential to Editor” section, and submit your "Accept" recommendation.

Reviewer #2: All comments have been addressed

Reviewer #3: (No Response)

2. Is the manuscript technically sound, and do the data support the conclusions?

Reviewer #2: Yes

Reviewer #3: Yes

3. Has the statistical analysis been performed appropriately and rigorously? 

Reviewer #2: N/A

Reviewer #3: N/A

4. Have the authors made all data underlying the findings in their manuscript fully available?

Reviewer #2: Yes

Reviewer #3: Yes

5. Is the manuscript presented in an intelligible fashion and written in standard English?

Reviewer #2: Yes

Reviewer #3: Yes

6. Review Comments to the Author

Reviewer #2: The manuscript is EXCELLENT and it was an absolute privilege to work this this esteemed group of authors to conduct the review. Thank you. The manuscript is now much more pleasurable to read and much easier to follow, well done :) The research itself is fascinating and comprises an amazing contribution to the literature.

Overall I have very few recommended revisions and feel that the manuscript is ready for publication once the items below are attended to.

1. On p. 7 (manuscript), line # 2 it would be worthwhile to be explicit again about how or why the researchers were trained/well-positioned to facilitate the critical reflections among clinicians. I know that earlier in the paper the authors comment on their suitability for dialogical exchanges (which was an excellent change, thank you!), but in the methods section please include a statement on what specifically qualified the researchers to promote critical reflexivity among clinicians. Otherwise, the section still presents as somewhat paternalistic and hierarchical (I know this is not actually the case but concerned that this is how it reads) (e.g., even if a statement could be made about any researchers that have focused specifically on critical reflexivity in practice). This is especially important given the findings related to how vulnerable the process of critical reflexivity might have felt to some participants (clinicians).

2. Please insert a reference at p. 8 (manuscript page) line #13 "Congruent with a realist approach, we analysed numerous data sources...". This requires a citation from the realist literature suggesting that multiple data sources should be included.

3. On p. 10 (manuscript), line #12, please include method for formal coding (i.e., NVivo, Dedoose, Word, Excel)

4. On p. 34 (manuscript), line #5/#6 states that the intervention could be used by other researchers, clinicians, or service designers to facilitate a shift. I think it is fair to say that other researcher could implement a similar intervention, potentially other service designers, but having been a clinician in a small non-research focused CTC I think it is unrealistic (for several reasons) to suggest that clinicians can or should implement this type of intervention. Instead, I wonder if the authors are planning to create any resources that could be disseminated to clinicians and more easily taken up in practice to at least educate/prompt regarding the human aspects of care (keeping in mind that many rehabilitation professionals in Canada and internationally do not have the luxury of working at large-research focused institutions). In fairness this could also be addressed in the discussion b/c issues with practice change were indicated in the introduction (i.e., how resource intensive it was to cultivate this shift in practice and the subsequent barriers to most rehabilitation professionals in taking up such a practice change, and what some tangible next steps could be to overcome barriers).

Reviewer #3: Thank you for the opportunity to review this manuscript. I found it to be methodologically innovative and sound. In addition, I believe the findings have broad applicability and transferability beyond MD clinics to interprofessional team-based care in a variety of healthcare contexts (e.g., both hospital and primary care).

I believe the authors did an excellent job responding to editorial reviews. Thank you for your attention to this. The paper has now been greatly strengthened with the inclusion of additional clarification around the human aspects of care, and detail regarding the approach and intervention (RE and nature of the dialogues). I also found the revisions increased the accessibility of the findings to the reader.

I have two minor comments for consideration:

1) Introduction (Pg. 4. Lines 9-12).

I had two re-read this sentence a couple of times to ensure that I understood that the authors were referring to interventions specifically designed to change clinical practice vs. interventions that clinicians engage in with patients. I often default to thinking of clinical vs. research interventions so perhaps that was the challenge for me. I wonder if a slightly rephrasing may increase clarification (e.g., As a result, it is not surprising that there is a general agreement in the literature that interventions designed to change clinician practice need to be sustained, repeated, and be contextually relevant).

2) Results (Pg. 25. Line 17). Ages are included in all other references to patients. Can you please add an age to Casey in this reference to ensure consistency?

I am recommending the paper be accepted for publication.

7. PLOS authors have the option to publish the peer review history of their article (what does this mean?). If published, this will include your full peer review and any attached files.

Reviewer #2: No

Reviewer #3: No

---

## [Editor Report · Acceptance letter]

18 Feb 2022

PONE-D-20-30857R1 

Enhancing human aspects of care with young people with muscular dystrophy: An evaluation of a participatory qualitative study with clinicians. 

Dear Dr. Setchell:

I'm pleased to inform you that your manuscript has been deemed suitable for publication in PLOS ONE. Congratulations! Your manuscript is now with our production department. 

Kind regards, 

on behalf of

Dr. Ukachukwu Okoroafor Abaraogu 

Academic Editor

PLOS ONE